# Unraveling the Shift of Visual Information Flow in MLLMs: From phased Interaction to Efficient Inference

## Abstract

Multimodal large language models (MLLMs) improve performance on vision-language tasks by integrating visual features from pre-trained vision encoders into large language models (LLMs). However, how MLLMs process and utilize visual information remains unclear. In this paper, a shift in the dominant flow of visual information is uncovered: (1) in shallow layers, strong interactions are observed between image tokens and instruction tokens, where most visual information is injected into instruction tokens to form cross-modal semantic representations; (2) in deeper layers, image tokens primarily interact with each other, aggregating the remaining visual information to optimize semantic representations within the visual modality. Based on these insights, we propose Hierarchical Modality-Aware Pruning (HiMAP), a plug-and-play inference acceleration method that dynamically prunes image tokens at specific layers, reducing computational costs by approximately 65% without sacrificing performance. Our findings offer a new understanding of visual information processing in MLLMs and provide a state-of-the-art solution for efficient inference. Code is released at https://anonymous.4open.science/r/HiMAP.

## 1 Introduction

In recent years, multimodal large language models (MLLMs) have emerged as an advanced architecture that integrates visual and textual information, demonstrating exceptional performance across various tasks (Liu et al., 2024c; Bai et al., 2023). Compared to traditional multimodal models, MLLMs achieve superior information fusion and complex semantic understanding by utilizing large language models (LLMs) (Brown, 2020) to process visual features (Liu et al., 2024a;b; Li et al., 2023a). However, the mechanisms of information interaction within these models remain underexplored. This study poses two critical questions: (1) To what extent do image tokens influence model

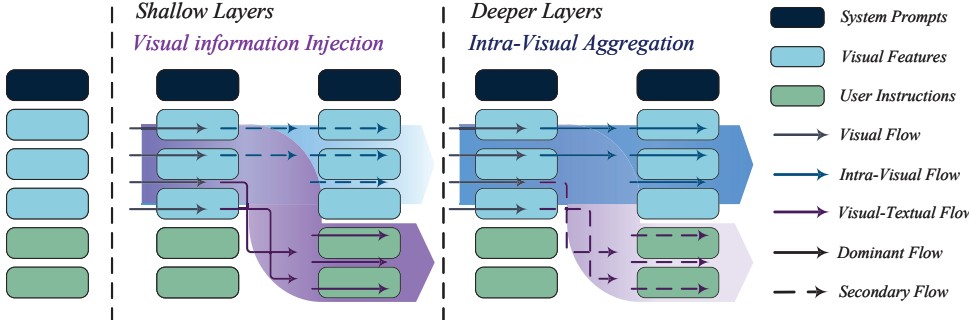

Figure 1: Illustration of our hypothesis. In shallow layers, image tokens inject most of the visual information into instruction tokens, establishing a cross-modal semantic representation for subsequent computations. In deeper layers, image tokens aggregate the residual visual information, refining the semantic representation within the visual modality.

predictions? (2) How is visual information processed within the model? These questions form the foundation of our investigation into the underlying working mechanism of MLLMs.

With respect to the first question, we developed three metrics based on saliency scores to quantify the impact of system tokens, image tokens, and instruction tokens on prediction outcomes. Experimental results indicated that the importance of image tokens was minimal, only equivalent to 0.03% of that of instruction tokens, despite image tokens comprising a significant portion of the model input. This limited impact can be attributed to two factors: (1) existing MLLMs fail to effectively learn visual features, and (2) visual signals exhibit considerable redundancy.

With respect to the second question, saliency analysis of the attention matrices reveals a strong interaction between image tokens and instruction tokens in shallow layers, while interactions among image tokens become more significant in deeper layers. This result intuitively reveals that as the model depth increases, the dominant flow of visual information within MLLMs undergoes a shift. Based on this, we propose the following hypothesis.

> *Phased Processing of Visual Information*
> $\mathcal{H}_1$: In shallow layers, image tokens primarily interact with instruction tokens, injecting most visual information into instruction tokens to establish a cross-modal semantic representation for subsequent computations.
> $\mathcal{H}_2$: In deeper layers, interactions among image tokens are enhanced, consolidating the residual visual information, thereby refining the semantic representation within visual modality.

Figure 1 provides a detailed elaboration of our hypothesis. Two experiments were designed to validate the aforementioned hypothesis. (1) By blocking the information interaction path between image and instruction tokens in certain layers, we observed such perturbations in shallow layers significantly degraded model performance, confirming that image tokens inject visual information into instruction tokens. (2) We compared the significance of visual-textual information flow and intra-visual information flow at various model depths, discovering that perturbations to intra-visual information flow in deeper layers led to more pronounced prediction deviations, thereby validating the interaction among image tokens to aggregate visual information. In summary, these results support our hypothesis, indicating that MLLMs process visual information differently at varying depths.

Despite their substantial computational cost, image tokens contribute minimally to prediction results. To address this issue, we propose a method for pruning image tokens to accelerate inference. Based on insights into internal information interactions within MLLMs, we introduce Hierarchical Modality-Aware Pruning (HiMAP), a plug-and-play technique that effectively streamlines the computational process by focusing the model on the most influential image tokens. HiMAP dynamically ranks the importance of image tokens according to the dominant visual information flow at different depths and applies pruning strategies at specified layers. By reducing the computational overhead of both self-attention modules and feed-forward networks modules, HiMAP reduce FLOPs by over 65%. Experimental results demonstrate that HiMAP can reduce inference latency by about 50% while maintaining model performance, rendering it nearly a "free lunch".

In summary, our contributions are fourfold: (1) identifying and analyzing the phenomenon where image tokens contribute minimally to prediction results in MLLMs; (2) uncovering latent patterns in the interaction between visual and textual information within MLLMs; (3) proposing HiMAP, a plug-and-play technique designed to reduce inference latency in MLLMs without compromising performance, leveraging insights from information interaction mechanisms; (4) Demonstrating the effectiveness of HiMAP across multiple vision-language tasks.

## 2    INEFFICIENT CONTRIBUTION OF IMAGE TOKENS

This section aims to illustrate the limited contribution of image tokens to model predictions. Section 2.1 provides an overview of the three token categories utilized in the inputs of MLLMs and their respective processing mechanisms. Section 2.2 analyzes the contributions of different modalities to prediction outcomes, using saliency score-based metrics. The quantitative analysis indicates that the visual modality contributes significantly less than its counterparts.

## 2.1 PRELIMINARIES

This section introduces how MLLMs process different tokens when generating output. Typically, these models follow a transformer decoder architecture (Vaswani et al., 2017), predicting responses autoregressively based on a given image-question pair.

Before being fed into the transformer decoder, multimodal information (including images and text) is converted into sequence embeddings. For images, a common approach involves extracting visual features using pre-trained encoders, such as CLIP-VIT (Radford et al., 2021). To align the dimensions of these visual features with the embedding size of LLMs and ensure semantic consistency, additional linear transformations or cross-attention modules are introduced. For text, natural language is tokenized into discrete units, and corresponding text embeddings are generated through embedding lookup. In this paper, "image tokens" and "text tokens" refer to both the discrete units of visual and textual data as well as the embeddings derived from them.

After preprocessing the image and text tokens into a unified embedding space, these tokens are input into the transformer decoder to generate output tokens. During this decoding process, the input tokens are categorized into three types: (1) **System Prompts**, which provide general information for controlling the behavior of MLLMs; (2) **Image Tokens**, derived from features learned by pre-trained visual encoders; and (3) **User Instructions**, which specify requests or questions related to the given images. The index sets of system, image, and instruction tokens are denoted by $\mathcal{S}$, $\mathcal{V}$, and $\mathcal{I}$, respectively. Comprehensive index set of all input tokens is represented as $\mathcal{X}$, where $\mathcal{X} = \mathcal{S} \cup \mathcal{V} \cup \mathcal{I}$.

A figure presenting the distribution of sequence lengths for the three types of input tokens are available in Appendix A. The sequence length for image tokens is 576, nearly twice the combined length of system and instruction tokens. This indicates that the computational load associated with image tokens in MLLMs is relatively higher.

## 2.2 VISUAL MODALITY IMPACT ASSESSMENT

This subsection quantitatively evaluates the impact of the visual modality on prediction outcomes. We employ the saliency technique (Wang et al., 2023; Simonyan, 2013), a widely used interpretability tool, to highlight key token interactions within the attention mechanism. Following established practices, we utilize Taylor expansion (Michel et al., 2019) to compute saliency scores for each element of the attention matrix:

$$I_l = \left| \sum_h A_{h,l} \odot \frac{\partial \mathcal{L}(x)}{\partial A_{h,l}} \right|. \tag{1}$$

Here, $A_{h,l}$ represents the attention matrix value for the $h$-th attention head in the $l$-th layer, $x$ denotes the input, and $\mathcal{L}(x)$ is the loss function of the task, e.g., the cross-entropy objective for question-answering tasks. The saliency matrix $I_l$ for the $l$-th layer is obtained by averaging across all attention heads. The significance of information flow from the $j$-th token to the $i$-th token in MLLMs is represented by $I_l(i, j)$. To illustrate the contributions of different modalities to prediction outcomes, three quantitative metrics based on $I_l$ are introduced:

$S_{sys}$, **which measures the importance of information flow from system tokens to other tokens:**

$$S_{sys} = \frac{1}{|\mathcal{S}|} \sum_{i \in \mathcal{X}} \sum_{j \in \mathcal{S}} I_l(i, j) \tag{2}$$

$S_{img}$, **which measures the importance of information flow from image tokens to other tokens:**

$$S_{img} = \frac{1}{|\mathcal{V}|} \sum_{i \in \mathcal{X}} \sum_{j \in \mathcal{V}} I_l(i, j) \tag{3}$$

$S_{ins}$, **which measures the importance of information flow from instruction tokens to other tokens:**

$$S_{ins} = \frac{1}{|\mathcal{I}|} \sum_{i \in \mathcal{X}} \sum_{j \in \mathcal{I}} I_l(i, j) \tag{4}$$

These three metrics enable a systematic observation of the information flow intensity from different modalities across various layers, facilitating the evaluation of their contributions to prediction outcomes.

We conducted experiments using the LLaVA-v1.5 series models on A-OKVQA and Sci-VQA datasets. Detailed experimental settings are outlined in Appendix B. Figure 2 demonstrates the impact of various modalities on prediction outcomes. Additional experimental results are available in Appendix C. The quantitative results reveal two key insights: (1) As the model depth exceeds a certain threshold, the influence of tokens on prediction outcomes decreases; (2) Instruction tokens exert the most significant effect on prediction outcomes, while image tokens have relatively little influence.

The first finding suggests that models do not efficiently utilize information in their final layers. The second finding reveals that the contribution of visual modality to prediction outcomes is relatively low. This limited impact of image tokens may be attributed to two main factors: (1) Existing MLLMs struggle to effectively learn visual features; and (2) there is substantial redundancy in the image signals fed into the models. This paper focuses on the latter issue, aiming to enhance the performance of MLLMs by reducing redundant information.

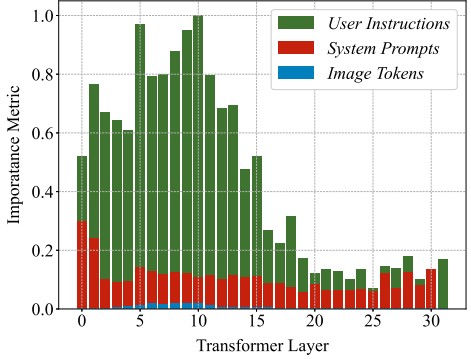 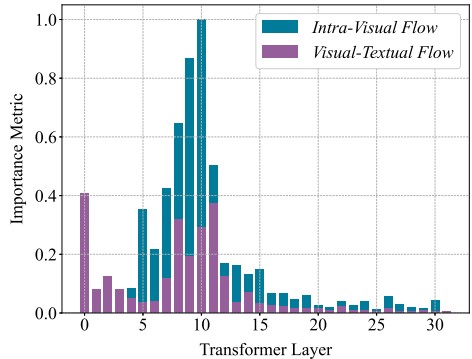

Figure 2: Contributions of different modalities to prediction outcomes across layers. The contribution of visual modality is significantly lower than textual modality.

Figure 3: Importance of intra-visual flow and visual-textual flow across layers. Dominant flow of visual information shifts as model depth increases.

## 3 SHIFT IN DOMINANT FLOW OF VISUAL INFORMATION

This section provides a comprehensive analysis of how MLLMs process visual information. In Section 3.1, two importance metrics are introduced to provide an intuitive understanding of visual information flow within MLLMs. The quantitative results lead to the following hypothesis: $\mathcal{H}_1$: In shallow layers, image tokens injects most visual information into instruction tokens to establish a cross-modal semantic representation for subsequent computations. $\mathcal{H}_2$: In deeper layers, image tokens aggregate the residual visual information, thereby refining the semantic representation within the visual modality. In Sections 3.2 and 3.3, we validate the hypothesis through information flow perturbation experiments.

### 3.1 HYPOTHESIS DRIVEN BY SALIENCY SCORES

This subsection seeks to uncover the underlying patterns of visual information interaction through attention mechanism in MLLMs. We continue to use $I_l(i, j)$ from Equation (1) to represent the significance of information flow from the $j$-th token to the $i$-th token. To clarify the visual information flow in MLLMs, we introduce two new quantitative metrics based on $I_l(i, j)$, with a particular focus on the information interaction involving image tokens. The metrics are defined as follows.

$S_{vv}$, measuring the importance of information flow among image tokens:

$$S_{vv} = \frac{1}{|\mathcal{V}|} \sum_{j \in \mathcal{V}} \sum_{i \in \mathcal{V}} I_l(i, j) \tag{5}$$

$S_{vt}$, measuring the importance of information flow from image tokens to instruction tokens:

$$S_{vt} = \frac{1}{|\mathcal{V}|} \sum_{j \in \mathcal{I}} \sum_{i \in \mathcal{V}} I_l(i, j) \tag{6}$$

$S_{vv}$ and $S_{vt}$ are utilized to analyze the mechanisms of visual information processing in MLLMs. Specifically, $S_{vt}$ quantifies the extent of information injection from image tokens to instruction tokens, whereas $S_{vv}$ measures the degree of information aggregation among image tokens. We define attention interactions among image tokens as intra-visual information flow and those between image and instruction tokens as visual-textual information flow.

**Results and Analysis** Figure 3 illustrates the rapid shifts in the significance of two information flows across different model depths. (1) In shallow layers (i.e., layers 1-3), the importance of the visual-textual information flow ($S_{vt}$) is notably high, while intra-visual information flow ($S_{vv}$) is comparatively low. (2) In deeper layers (i.e., layers 8-16), the intra-visual information flow ($S_{vv}$) becomes dominant. Additional experimental results are available in Appendix C.

**Proposed Hypothesis** Based on observations of shifts in the dominant flow of visual information, we propose a hypothesis concerning the phased processing of visual information in MLLMs. In shallow layers, image tokens primarily interact with instruction tokens, injecting most visual information into these tokens to establish a cross-modal semantic representation for subsequent computations. In deeper layers, interactions between image tokens intensify, consolidating the residual visual information, thereby refining the semantic representation within the visual modality. Figure 1 illustrates this hypothesis in detail.

### 3.2 SHALLOW LAYERS: VISUAL INFORMATION INJECTION

In this section, we validate the first part of our hypothesis. We propose that injecting visual information into instruction tokens depends on the the information flow from image tokens to instruction tokens, facilitated by the attention mechanism. By manipulating attention layers and disrupting the visual-textual information flow, we aim to confirm the presence of this injection process and its effect on prediction outcomes.

**Implementation Details** To disrupt the visual-textual information flow, we block the interaction between image and instruction tokens by adjusting the attention matrix $A$. Specifically, we set $A_l(i, j)$ to 0 for $i \in \mathcal{I}$ and $j \in \mathcal{V}$ in the attention matrix $A_l$ of the $l$-th layer. This modification prevents the instruction tokens from receiving information from the image tokens in the $l$-th layer.

**Evaluation Metrics** Inspired by the loyalty metrics proposed by Wang et al. (2023), we design the following metrics to assess the impact of disrupting visual-textual information flow: **(1) Label Consistency**: evaluates how consistent the prediction outcomes are before and after disruption. **(2) Score Consistency**: applies the Jaccard similarity to compare the top-5 predicted tokens before and after disruption, capturing broader changes in prediction results. A lower consistency score indicates a greater impact on prediction outcomes.

**Results and Analysis** We conducted experiments with the LLaVA-v1.5 series models on Sci-VQA and AOKVQA datasets, with detailed experimental setups provided in Appendix B. As shown in Figure 4, model performance significantly deteriorated after disruptions in the first five layers, but this effect weakened as network depth increased. In contrast, disrupting the information flow from image tokens to random tokens had a relatively minor impact on performance. These findings support the conclusion that visual information is injected into instruction tokens in shallow layers.

### 3.3 DEEPER LAYERS: INTRA-VISUAL AGGREGATION

This section further validates the second part of the hypothesis, which posits that in deeper layers, enhanced interactions between image tokens lead to aggregation of residual visual information. To

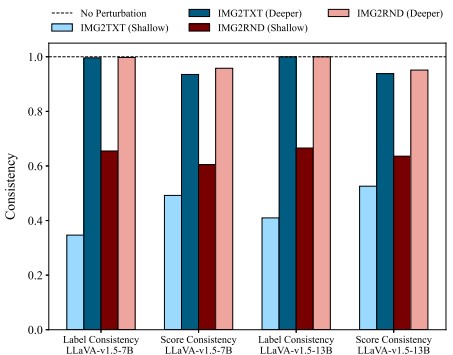 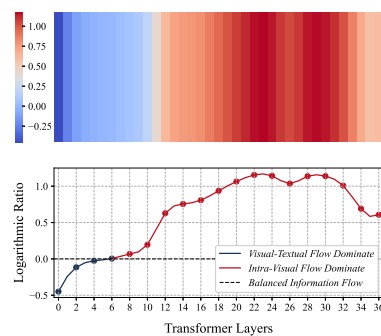

Figure 4: Impact of disrupting visual-textual flow versus disrupting visual-random flow within the first or last 5 layers. Disrupting visual-textual flow in the first 5 layers has the most substantial effect, highlighting the importance of shallow-layer information injection from image tokens to instruction tokens.

Figure 5: The values of $D_l$ for every two layers in LLaVA-v1.5-13B. In deeper layers, $D_l > 0$, indicating that disruptions in intra-visual flow lead to greater prediction biases, thus validating the aggregation of residual visual information through interactions between image tokens.

investigate this, we manipulated the attention layers to separately disrupt the intra-visual and visual-textual information flows. By comparing the effects of these two disruptions on prediction outcomes, we confirm changes in the underlying visual processing mechanisms.

**Implementation Details** We modified the attention matrix $A$ to block interactions between image tokens, thereby disrupting intra-visual information flow. Specifically, we set $A_l(i, j)$ to 0 for $i, j \in \mathcal{V}$ in the attention matrix $A_l$ of the $l$-th layer, thereby preventing information interactions among image tokens within that layer. The disruption in visual-textual information flow is consistent with the procedure described in section 3.2.

**Evaluation Metrics** The prediction biases resulting from disruptions in the visual-textual and intra-visual information flows are denoted as $E_{vt,l}$ and $E_{vv,l}$, respectively, where $l$ refers to the $l$-th layer. To quantify the relative impact of these two disruptions on prediction outcomes, we introduce $D_l$:

$$D_l = \log(\frac{E_{vv,l}}{E_{vt,l}}). \tag{7}$$

This metric represents the logarithmic ratio of prediction biases caused by two distinct disruptions in visual information flow. When $D_l > 0$, it indicates that intra-visual information flow dominates in the $l$-th layer. Conversely, when $D_l < 0$, it suggests that visual-textual information flow prevails in the $l$-th layer. We provide more implementation details about this experiment in Appendix D, as well as the reasons for using $D_l$ as the evaluation metric.

**Results and Analysis** We conducted experiments using the LLaVA-v1.5-13B model on Sci-VQA and A-OKVQA datasets, with the experimental setup detailed in Appendix D. Figure 5 presents the experimental results, which have been averaged across both datasets. In deeper layers, the value of $D_l$ approaches 1.2, indicating the aggregation of residual visual information. In contrast, the value of $D_l$ drops to -0.5 in shallow layers, further supporting the injection of visual information from image tokens into instruction tokens.

## 3.4 HYPOTHESIS DISCUSSION

In section 3.2, we validate that image tokens inject most visual information into instruction tokens in shallow layers. In section 3.3, we validate that image tokens aggregate the residual visual information in deeper layers. As discussed in Section 2.2, the contribution of image tokens to prediction outcomes is inefficient. We posit that not all image tokens are necessary at every layer. In shallow layers, Once most visual information has been injected into the instruction tokens, many image to-

kens lose significance. This explains why visual-textual information flow exerts minimal influence on prediction outcomes in subsequent layers. In deeper layers, after the residual visual information is aggregated, most of the remaining image tokens become redundant. These insights support the development of a hierarchical token-pruning algorithm aimed at accelerating the inference process.

# 4 HIERARCHICAL MODALITY-AWARE PRUNING

Given the relatively minor contribution of image tokens to prediction outcomes, coupled with their substantial computational cost in MLLMs, we propose a dynamic pruning method for image tokens. This approach effectively reduces computational overhead during inference without compromising model performance.

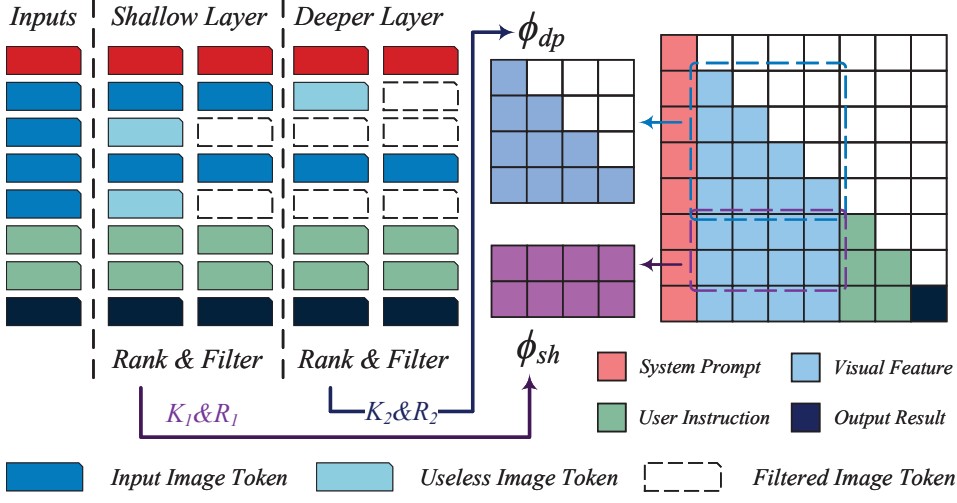

Figure 6: Illustration of Hierarchical Modality-Aware Pruning (HiMAP). In shallow layers, HiMAP ranks image tokens at the $K_1$-th layer based on the importance criterion $\phi_{sh}$, removing the tokens in the bottom $R_1\%$. In deeper layers, HiMAP ranks the remaining image tokens at the $K_2$ layer according to the importance criterion $\phi_{dp}$, filtering out those in the bottom $R_2\%$.

## 4.1 HIERARCHICAL IMAGE TOKEN PRUNING

Given the phased processing mechanism of visual information in MLLMs, we propose Hierarchical Modality-Aware Pruning (HiMAP), a technique that accelerates the inference process by dynamically pruning image tokens.

Figure 6 illustrates the overall framework of HiMAP, which includes two core components: shallow-layer pruning module and deeper-layer pruning module. Each module features an importance ranking function $f_\phi$ and two parameters: the filtering layer $K$ and the filtering ratio $R\%$. At the $K$-th layer of MLLMs, the ranking function $f_\phi$ accepts a set of image tokens as input and ranks them according to a predefined importance criterion $\phi$. After ranking, image tokens deemed to have the lowest importance in the bottom $R\%$ are pruned in subsequent layers, thus optimizing the utilization of computational resources.

In shallow layers, image tokens primarily interact with instruction tokens, injecting most visual information into instruction tokens. Consequently, we define the importance criterion $\phi_{sh}$ in shallow-layer pruning module as the sum of attention scores from the given image token $v$ to all instruction tokens, represented as:

$$\phi_{sh}(v) = \sum_{i \in \mathcal{I}} A_{K_1}(i, v). \tag{8}$$

Here, $K_1$ denotes the filtering layer in shallow-layer pruning module. This criterion quantifies the influence of image tokens on instruction tokens, thereby guiding the pruning of image tokens in shallow layers.

In deeper layers, interactions among image tokens are enhanced, consolidating the residual visual information. Thus, we define the importance criterion $\phi_{dp}$ in the deeper-layer pruning module as the sum of attention scores from all other image tokens to the given image token $v$, expressed as:

$$\phi_{dp}(v) = \sum_{i \in \mathcal{V}} A_{K_2}(v, i) \tag{9}$$

Here, $K_2$ denotes the filtering layer in deeper-layer pruning module. This criterion evaluates the information interaction between image tokens, directing the pruning of image tokens in deeper layers.

## 4.2 COMPUTATION COST ESTIMATION

We estimate the computational cost of the multi-head attention (MHA) and feed-forward neural network (FFN) modules in terms of Floating Point Operations Per Second (FLOPs). For a transformer layer, given that the input contains $n$ image tokens, the hidden layer dimension is $d$, and the intermediate layer dimension of the FFN is $m$, the FLOPs for this layer can be represented as $\Omega(n) = 4nd^2 + 2n^2d + 2ndm$.

For the entire model, assuming there are $L$ layers in total, the shallow-layer pruning module reduces the number of image tokens from $n$ to $n_1 = (1 - R_1\%) \cdot n$ at the $K_1$-th layer. The deeper-layer pruning module further reduces the number of image tokens to $n_2 = (1 - R_2\%) \cdot n_1$. Thus, the theoretical FLOPs reduction rate $\eta$ related to image tokens can be calculated using the following formula:

$$\eta = 1 - \frac{K_1 \cdot \Omega(n) + (K_2 - K_1) \cdot \Omega(n_1) + (L - K_2) \cdot \Omega(n_2)}{L \cdot \Omega(n)} \tag{10}$$

## 5 EXPERIMENTS

This section presents the experimental results of HiMAP across various tasks. Subsection 5.1 outlines the different tasks used to evaluate HiMAP. Subsection 5.2 provides quantitative results demonstrating that HiMAP significantly reduces computational costs while maintaining model performance.

Table 1: Performance and computational cost of HiMAP on the multiple-choice QA task & object hallucination task with **highest** score for each model highlighted in **red** and the **lowest** computational cost in **green**.

| Model | Method | TFLOPs | Ratio | Sci-VQA | A-OKVQA | POPE |
|---|---|---|---|---|---|---|
| LLaVA-v1.5-7B | Baseline | 2.98 | 100% | **67.9** | 76.6 | **86.4** |
| | FastV | 1.56 | 54% | 68.1 | **77** | 84.9 |
| | HiMAP | **0.73** | **24%** | **68.3** | **77.2** | 86.2 |
| LLaVA-v1.5-13B | Baseline | 5.81 | 100% | **71.6** | **82** | **87.2** |
| | FastV | 3.09 | 53% | 71.3 | 81.3 | 84.8 |
| | HiMAP | **1.36** | **23%** | **72.1** | **81.4** | 86.5 |
| QwenVL-Chat-7B | Baseline | 3.6 | 100% | 68 | **75.7** | **84.5** |
| | FastV | 1.9 | 53% | **68.2** | 75.3 | 82.7 |
| | HiMAP | **0.89** | **25%** | **68.5** | **75.9** | 83.7 |

## 5.1 EVALUATION TASKS

We conducted a comprehensive evaluation of HiMAP across multiple vision-language tasks to ensure that model performance remained consistent across various domains. Four distinct categories of vision-language tasks were selected for this evaluation, as detailed below:

**Multiple-Choice Question Answering.** This task requires the model to select the correct answer from predefined options based on a given image and accompanying question. We utilized

Table 2: Performance and computational cost of HiMAP on the image caption task & open-ended QA task with **highest** score for each model highlighted in **red** and the **lowest** computational cost in **green**.

| Model | Method | TFLOPs | Ratio | Nocaps | Flickr30k |
|---|---|---|---|---|---|
| LLaVA-v1.5-7B | Baseline | 2.98 | 100% | **78.8** | **50.9** |
| | FastV | 1.56 | 54% | 78.6 | 50.6 |
| | HiMAP | **1.01** | **34%** | **78.7** | **51.3** |
| LLaVA-v1.5-13B | Baseline | 5.81 | 100% | 82.8 | 53.6 |
| | FastV | 3.06 | 53% | **82.9** | **53.8** |
| | HiMAP | **1.93** | **33%** | **83.7** | **53.8** |
| Model | Method | TFLOPs | Ratio | LLaVA-Bench | MM-Vet |
| LLaVA-v1.5-7B | Baseline | 2.98 | 100% | **65.7** | **33.4** |
| | FastV | 1.56 | 54% | 62.4 | 31.2 |
| | HiMAP | **1.01** | **34%** | **66.5** | **33.7** |
| LLaVA-v1.5-13B | Baseline | 5.81 | 100% | **73.5** | **37.4** |
| | FastV | 3.06 | 53% | 71.7 | 35.5 |
| | HiMAP | **1.93** | **33%** | **74.5** | **37.3** |

A-OKVQA (Schwenk et al., 2022) and Sci-VQA datasets (Goyal et al., 2017) as benchmarks, employing accuracy as the evaluation metric.

**Image Captioning.** This task involves generating a concise description of a given image. We employed NoCaps and Flickr30k datasets (Agrawal et al., 2019; Plummer et al., 2015) for benchmarking, using the CIDEr score (Vedantam et al., 2015) as the evaluation metric.

**Open-ended Question Answering.** This task requires the model to generate a relevant response based on a provided image and query. We utilized LLaVA-Bench and MM-Vet datasets (Yu et al., 2023) for evaluation, with GPT-4o (Achiam et al., 2023) assessing the quality of generated responses.

**Object Hallucination.** This task evaluates the ability of the model to accurately determine the presence of specific objects within a given image, thereby assessing the reliability of the generated responses. The POPE dataset (Li et al., 2023b) served as the benchmark, with accuracy employed as the evaluation metric.

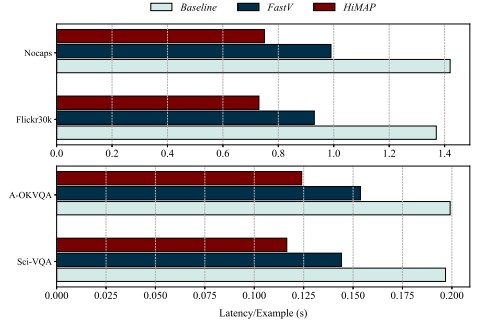
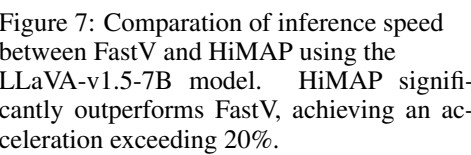
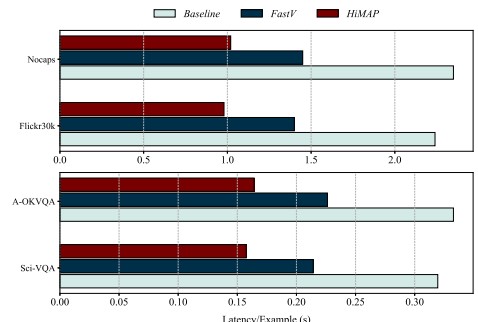

Figure 7: Comparison of inference speed between FastV and HiMAP using the LLaVA-v1.5-7B model. HiMAP significantly outperforms FastV, achieving an acceleration exceeding 20%.

Figure 8: Comparison of inference speed between FastV and HiMAP using the LLaVA-v1.5-13B model. Remarkably, HiMAP enables a 13B model to achieve faster inference than a 7B model.

## 5.2 RESULTS AND ANALYSIS

We primarily compared our method with the FastV method (Chen et al., 2024), utilizing its optimal configuration of $K = 2$ and $R = 50\%$. The experimental results demonstrate that HiMAP can reduce more computational costs while having a smaller comprise on model performance.

**Analysis of Impact on Model Performance.** Table 1 illustrates the effects of our method and FastV method on model performance in multiple-choice question-answering and object hallucination tasks. In most multiple-choice question-answering scenarios, our method not only maintains but also enhances model performance. For instance, on Sci-VQA dataset, all three models demonstrate performance improvements after the application of our method, which we attribute to the reduction of redundant computations that yield additional gains. In the object hallucination task, our method results in a decline of approximately 0.5%; however, relative to FastV method, our approach brings lesser negative impact. Table 2 provides an overview of the performance of our method and FastV method on image description and open-ended question-answering tasks. In contrast to the general performance decline of approximately 1.5% associated with FastV method, our method contributes to an improvement in model performance.

**Analysis of Computational Cost.** Tables 1 and 2 illustrate the substantial reduction in computational costs attained by our method. With parameters set at $K_1 = 2, R_1 = 50\%, K_2 = 8$, and $R_2 = 75\%$, our approach reduces FLOPs by 75%, significantly surpassing the 45% reduction achieved by FastV method. Figures 7 and 8 compare the actual inference speeds of our method against FastV method, demonstrating that our approach provides an additional 20% improvement.

## 6 RELATED WORK

**Interpretability of LLMs.** Research on attention mechanisms has significantly enhanced our understanding of large language models. For instance, Xiao et al. (2023) highlight a phenomenon known as *attention sink*, indicating that maintaining the key-value states of initial tokens can largely restore the performance of window attention (Song et al., 2022), primarily due to the strong attention scores associated with these tokens. Furthermore, Wang et al. (2023) discovered that label words serve as anchors in in-context learning, facilitating the aggregation and distribution of task-relevant information. In addition, Wu et al. (2024) identified a specific category of attention heads, referred to as retrieval heads, which are primarily responsible for extracting relevant information from lengthy contexts. However, most studies on attention mechanisms focus exclusively on text-based models, creating a gap in our understanding of information interaction within MLLMs. Our research aims to bridge this gap, offering new insights into how MLLMs process and utilize visual information.

**Inference Optimization for LLMs.** Research on efficient inference in large language models has primarily focused on two categories of optimization: (1) Memory Consumption Optimization, which includes methods such as FlashAttention (Dao et al., 2022), vLLM (Kwon et al., 2023), and RingAttention Liu et al. (2023) that enhance the memory efficiency of the attention module without significantly altering outcomes; and (2) Computation Simplification, which involves techniques like StreamingLLM and FastGen (Holmes et al., 2024) that improve inference efficiency by eliminating redundant attention calculations. This paper emphasizes the latter category. Most existing methods target text-only models, creating a notable gap in their applicability to MLLMs. Recent strategies, including FastV and VTW (Lin et al., 2024), have accelerated inference speeds through image token pruning, yet they overlook the shift in the dominant flow of visual information, failing to fully harness the potential for accelerating the inference of MLLMs.

## 7 CONCLUSION

In this paper, we propose a hypothesis regarding visual information processing in MLLMs, suggesting that image tokens inject most visual information into instruction tokens in shallow layers while consolidate the remaining visual information in deeper layers. Results from information flow perturbation experiments confirm this hypothesis for the LLaVA-v1.5 series models. Building on these insights, we introduce Hierarchical Modality-Aware Pruning, a plug-and-play method that dynamically prunes image tokens at specific layers to improve inference speed. This method not only reaffirms our hypothesis but also demonstrates significant potential for practical applications.

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

## A  DISTRIBUTION OF SEQUENCE LENGTHS

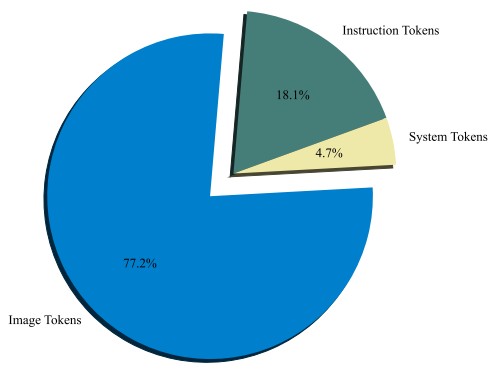

Figure 9: Distribution of sequence lengths

## B  DETAILED EXPERIMENTAL SETTING FOR ANALYSIS EXPERIMENTS

The analysis experiments on visual information flow were conducted on the A-OKVQA and Sci-VQA datasets. We filtered samples from both datasets that contained valid visual information. Specifically, we removed the image information from the multimodal samples and observed whether the model could still predict the correct answer based solely on the text. If the model could still predict the correct answer without visual information, it indicated that the sample did not contain valid visual information, and we discarded these pseudo-multimodal samples. We ran the analysis experiments on a server equipped with an 80G A800 GPU. Unless otherwise specified, the experimental results were averaged across the two datasets.

## C  ADDITIONAL RESULTS OF INFORMATION FLOW ANALYSIS

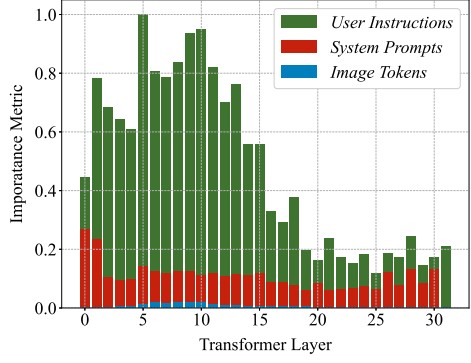

Figure 10: Contributions of different modalities to prediction outcomes across layers.(a-okvqa 7B)

Figure 11: Importance of intra-visual flow and visual-textual flow across layers.(a-okvqa 7B)

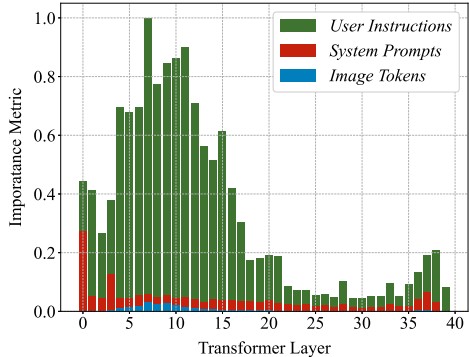

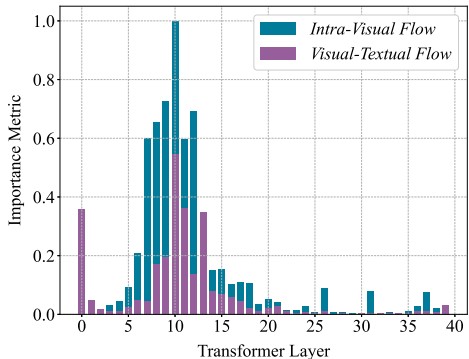

Figure 12: Contributions of different modalities to prediction outcomes across layers.(a-okvqa 13B)

Figure 13: Importance of intra-visual flow and visual-textual flow across layers.(a-okvqa 13B)

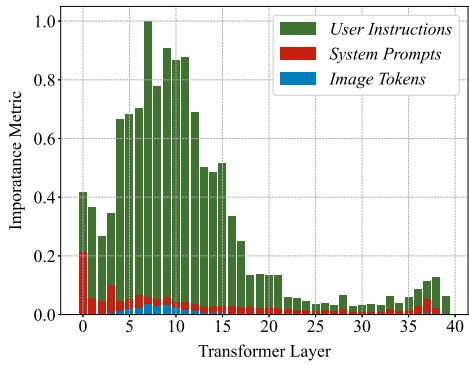

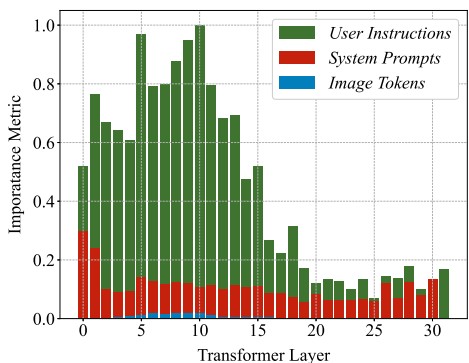

Figure 14: Contributions of different modalities to prediction outcomes across layers.(sci-vqa 13B)

Figure 15: Importance of intra-visual flow and visual-textual flow across layers.(sci-vqa 13B)

## D  DETAILS OF DISRUPTION EXPERIMENT IN DEEPER LAYER

**Explanation for $E_l$ .** When there is no disturbance to the visual information flow, **Score Consistency** is denoted as $S$, which is the number of correctly predicted samples. After disturbing the visual information flow, let the **Score Consistency** be $\hat{S}$, where $\hat{S} < S$. The calculation formula for $E_l$ is $E_l = S - \hat{S}$. We measure the error caused by the disturbance of the information flow through the change in **Score Consistency**.

**Explanation for $D_l$.** We use the $D_l$ metric to validate the importance of the intra-visual information flow, based on two main considerations: (1) as demonstrated by the experimental results in Section 2.2, the prediction outcomes are primarily influenced by intra-textual information flow, which weakens as the network depth increases. Consequently, although intra-visual information flow becomes more prominent in deeper layers, its disruption has minimal impact on prediction outcomes. Therefore, we use the significance of visual-textual information flow as a baseline and apply a logarithmic ratio to measure the variation in the importance of intra-visual information flow; (2) we focus on the relative strength between intra-visual and visual-textual information flows to clearly illustrate the shift in the mechanism of visual information processing.

# E HYPERPARAMETER SETTINGS OF HIMAP

**Model Configuration** We utilized LLaVA-v1.5-7B, LLaVA-v1.5-13B and QwenVL-Chat-7B as our evaluation models. To ensure the reproducibility of the experimental results, we employed a greedy search strategy for generation. A detailed list of the prompts used by the multimodal large language models for each task is provided as Following. All experiments were conducted on a server with one 80GB A800 GPU.

- **Image Caption.** For image captioning tasks Nocaps and Flickr30k, we adopt prompt as "Provide a one-sentence caption for the provided image."
- **A-OKVQA.** For A-OKVQA, we adopt the the multiple choice version of evaluation and use prompt as: "Analyse the image and choose the best answer for the following question: {question} Options: {options}. Output the letter of the correct answer."
- **Sci-VQA.** For Sci-VQA, we adopt the the multiple choice version of evaluation and use prompt as: "{question} Options: {options}. Answer with the option's letter from the given choices directly."
- **Object Hallucination.** For POPE, we adopt prompt as "question: {question} Please just answer yes or no."

Table 3: Hyperparameter settings of HiMAP for various models and vision-language tasks

| Model | Evaluation Task | HiMAP Configuration | | | |
|---|---|---|---|---|---|
| | | $K_1$ | $R_1$ | $K_2$ | $R_2$ |
| LLaVA-v1.5-7B | Multiple-Choice QA & Object Hallucination | 2 | 50% | 8 | 75% |
| | Image Caption & Open-ended QA | 2 | 50% | 15 | 75% |
| LLaVA-v1.5-13B | Multiple-Choice QA & Object Hallucination | 3 | 50% | 8 | 75% |
| | Image Caption & Open-ended QA | 3 | 50% | 15 | 75% |

**HiMAP Configuration** Based on the analysis of the importance of visual information flow in section 3.1, the hyperparameter settings for HiMAP across different tasks and models are presented in Table 3. For the multiple-choice question answering and object hallucination tasks, we adopted more aggressive parameter settings, significantly pruning image tokens to reduce computational costs. For the open-ended question answering and image captioning tasks, we used more conservative parameter settings to allow the model to access more visual information.

