# OpenReview forum: "Unraveling the Shift of Visual Information Flow in MLLMs: From Phased Interaction to Efficient Inference"
_ICLR.cc/2025/Conference — ICLR 2025 Conference Withdrawn Submission_

### Official Review · Reviewer_Vzxh · 2024-10-30

**Soundness:** 2
**Presentation:** 1
**Contribution:** 2
**Rating:** 3
**Confidence:** 4

**Summary:**

In this paper, authors begin by studying how visual tokens interact in MLLMs and observe that 1) image tokens interact strongly with text instruction tokens to form cross-modal representations in shallow layers; 2) image tokens aggregate remaining visual information in deeper layers. Upon this, they propose a token pruning inference strategy, HiMAP, for MLLM, by selecting the most important image tokens by image-text-attention scores in shallow layers and image-self-attention scores in deeper layers.

**Strengths:**

1) The paper is well motivated, authors design the HiMAP strategy corresponding to the behavior of how visual information is utilized in MLLM layers.

2) HiMAP manages to reduce the computational costs by approximately 65% without sacrificing performance.

**Weaknesses:**

1) The ablation studies are insufficient, e.g., different choices of K1, K2 , R1, R2;  ablation for using different importance criteria on shallow layers and deep layers.

2) The finding that LLMs may well process visual tokens in the early layers has already been proposed in previous works[1-3]. The stagewise token pruning strategy has also been proposed for efficient MLLM [3]. Consequently, the novelty of this paper is somewhat limited.

3) More benchmarks for MLLM performance evaluation should be involved to exhibit the effectiveness of HiMAP, e.g., the widely used GQA, MME, MM-Vet, VQAv2 and etc.
The paper should be drafted with line number on the page and the writing of the paper should be improved.


[1] An Image is Worth 1/2 Tokens After Layer 2: Plug-and-PLay Acceleration for VLLM Inference, in ECCV24.

[2] DeepStack: Deeply Stacking Visual Tokens is Surprisingly Simple and Effective for LMMs, in NeurIPS24.

[3] LLaVolta: Efficient Multi-modal Models via Stage-wise Visual Context Compression, in NeurIPS24.

**Questions:**

Please refer to weakness.

---

### Official Review · Reviewer_kQ13 · 2024-11-04

**Soundness:** 2
**Presentation:** 3
**Contribution:** 3
**Rating:** 6
**Confidence:** 3

**Summary:**

This paper proposes that in MLLMs, image tokens primarily convey visual information to instruction tokens in shallow layers, while deeper layers consolidate the remaining visual data. Based on this insight, a plug-and-play visual pruning method, HiMAP, is proposed to reduce the computational costs.

**Strengths:**

* From the novel perspective of information flow, the authors have analyzed the fusion patterns of visual features in MLLMs. Building upon this analysis, they proposed an adaptive approach for visual redundancy elimination.
* The structural design of HiMAP is intuitive and demonstrates robust performance across a range of tasks, exemplified by image captioning and VQA.
* The article is highly readable, featuring a well-defined and clear structure.

**Weaknesses:**

* In analyzing the information flow between visual tokens and textual tokens, it is essential to thoroughly examine the flow in both directions. Hypothesis H1 is valid only if it is determined that the primary flow of information occurs from the visual modality to the textual modality, rather than in the opposite direction. This requires conducting an experiment to compare the magnitudes of $S_{vt}$ in Equation 6 with $S_{tv}$.

* Furthermore, merely showing a decline in performance by restricting the interaction between image tokens and instruction tokens in shallow layers does not sufficiently support Hypothesis H1. It is essential to complement this with an experiment that specifically limits the flow of intra-visual information within shallow layers (the current IMG2RND experiment in Figure 4 is not direct). Only when the resulting performance degradation is considerably less pronounced can Hypothesis H1 be adequately substantiated.

**Questions:**

* Does the type of task potentially influence the conclusion regarding the minimal importance of visual tokens, which account for only 0.03% of the significance of textual tokens? For example, the proportion of visual tokens may considerably decrease in multi-turn dialogue tasks. At the same time, their relative significance could increase due to the normalization of sequence length reflected in Equations 2, 3, and 4.
* The insight presented in line 7 on page 4 looks weird.  If the contribution of tokens from deeper layers to response prediction is low, why not leverage tokens from the shallow layer with the most significant contribution to generating responses?  In this reviewer's opinion, only the comparison within each layer in Figure 2 carries practical significance.
* The reviewer suggests evaluating the performance of HiMAP against the baseline on fine-grained perception tasks, such as document understanding and OCR (e.g., Chartqa[1] and Docvqa[2]). This would provide a more solid demonstration of HiMAP's efficacy in reducing redundant image tokens.

[1] Chartqa: A benchmark for question answering about charts with visual and logical reasoning. \
[2] Docvqa: A dataset for vqa on document images.

---

### Official Review · Reviewer_Qbnk · 2024-11-05

**Soundness:** 1
**Presentation:** 3
**Contribution:** 2
**Rating:** 3
**Confidence:** 3

**Summary:**

This paper examines the significance of image tokens in different layers of MLLMs, suggesting that image tokens tend to facilitate modality injection in shallow layers and engage in more internal interactions in deeper layers. Based on this analysis, the paper proposes HiMAP, an algorithm for dynamically pruning image tokens to accelerate inference, which has been validated for its effectiveness across various multimodal tasks.

**Strengths:**

1. Universality: As a universal image token pruning algorithm, HiMAP can be easily applied to different architectures of MLLMs to achieve accelerated inference.
2. Usability: The method is straightforward, with low transfer costs.
3. The saliency scores and dynamic pruning approach used in the paper can provide inspiration for the field of accelerated inference.

**Weaknesses:**

1. The paper lacks line numbers, which seems to deviate from ICLR's submission standards and may hinder reviewers in accurately pinpointing issues within the document.
2. The experiments are not comprehensive enough, with validation only on a limited number of tasks. As a universal solution, it should be tested on common multimodal benchmarks, such as LLava-Bench, MMBench, etc.

**Questions:**

1. As mentioned in Weakness #1.
2. In Section 2.2, the authors derive two main factors based on insights 1) "As the model depth exceeds..." and 2) "Instruction tokens exert the most...". The first conclusion is undoubtedly correct, but the second remains questionable. Although much work has validated the redundancy of image tokens in MLLMs, the two insights provided in this paper do not directly lead to this conclusion. The "limited impact of image tokens" mentioned in the paper only supports the first conclusion, while the argument for the second conclusion would require a computation of saliency for each image token (assuming a length of N), and if the authors conducted this experiment, they would find that only some image tokens have high significance.

---

### Official Review · Reviewer_nG12 · 2024-11-06

**Soundness:** 3
**Presentation:** 3
**Contribution:** 3
**Rating:** 5
**Confidence:** 3

**Summary:**

This paper identifies the minimal role of image tokens in MLLM predictions, uncovers patterns in visual-textual interactions, introduces HiMAP as a pruning technique to reduce inference latency without compromising performance, and demonstrates its effectiveness across diverse vision-language tasks.

**Strengths:**

1. The paper is well-written, with clear explanations and helpful visuals.

2. This paper introduces intriguing hypotheses and includes extensive, detailed studies to support them.

3. Extensive experiments confirm HiMAP’s effectiveness, showing reduced computational costs while preserving performance.

**Weaknesses:**

1. Lack of ablation studies. For example, additional experiments could be included to examine the impact of various pruning strategies on the model and to assess the effects of different hyperparameter settings, such as K1, K2 and the ratio.

2. The authors might consider including additional benchmarks, such as MME and AI2D, and presenting fine-grained performance scores. Additionally, it would be helpful to include metrics such as GPU memory and total time in the comparisons to provide a more comprehensive evaluation.

**Questions:**

I noticed that VTM suggests that visual tokens are not essential in the deeper layers of MLLMs and strategically withdraw them at a certain layer. I'm very curious about the advantages of HiMAP compared to VTM.

---

### Official Review · Reviewer_yfzG · 2024-11-07

**Soundness:** 3
**Presentation:** 3
**Contribution:** 2
**Rating:** 5
**Confidence:** 4

**Summary:**

This paper dive into how multimodal large language models (MLLMs) process and utilize visual information. Based on the widely used saliency technique for interpretability, information flow among different tokens across different layers is analyzed. The authors find out that visual information injection dominates in shallow layers while intra-visual aggregation dominates in deeper layers. Finally, hierarchical image token pruning is proposed to prune at both shallow and deep layer with specific criterion.

**Strengths:**

1.	The paper is well-written and easy understanding. The figure 1 and 6 are intuitive to understand the overall framework.
2.	The saliency technique used for analyzing the information flow among various tokens are interesting and intuitive. The conclusion that visual information injection dominates in shallow layers while intra-visual aggregation dominates in deeper layers makes sense.
3.	Experimental results demonstrate the effectiveness of the proposed method to some extent.

**Weaknesses:**

1.	The phenomenon analyzed in the paper is not surprising, and previous works [1][2] have pointed out the similar findings that information of vision tokens has migrated to following text tokens within the first few layers of MLLMs. Thus, I think this paper is with limited novelty which employs a commonly used technique to analyze the phenomenon that have been identified.
2.	I wonder how the parameter K1, K2 is determined. For different datasets and tasks, the parameters may be different. Directly setting k1 and k2 to a pre-defined value may be not suitable. Could K1, K2 be dynamically adjusted based on the input samples?
3.	The evaluation datasets used in the paper is quite limited. I suggest the authors to evaluate on other commonly used datasets, especially OCR-related or fine-grained datasets to demonstrate the effectiveness, e.g., textvqa, gqa, docvqa, chartqa, seed-bench. For the efficiency evaluation, I suggest the authors to include inference time and GPU memory.
4.	Two different criteria are used in shallow and deep layers. I wonder the performance if the same criterion is used. If the performance is similar, the analysis of different information flow in shallow and deep layers is not very convincing.

[1] An Image is Worth 1/2 Tokens After Layer 2: Plug-and-PLay Acceleration for VLLM Inference.

[2] BOOSTING MULTIMODAL LARGE LANGUAGE MODELS WITH VISUAL TOKENS WITHDRAWAL FOR RAPID INFERENCE

**Questions:**

Please refer to the 'weaknesses' part.

---

### Note · Authors · 2024-11-15

I have read and agree with the venue's withdrawal policy on behalf of myself and my co-authors.